# Characteristics and Therapeutic Potential of Human Amnion-Derived Stem Cells

**DOI:** 10.3390/ijms22020970

**Published:** 2021-01-19

**Authors:** Quan-Wen Liu, Qi-Ming Huang, Han-You Wu, Guo-Si-Lang Zuo, Hao-Cheng Gu, Ke-Yu Deng, Hong-Bo Xin

**Affiliations:** 1The National Engineering Research Center for Bioengineering Drugs and the Technologies, Institute of Translational Medicine, Nanchang University, Nanchang 330031, China; liuquanwen@ncu.edu.cn (Q.-W.L.); hymeng2000@aliyun.com (Q.-M.H.); 6303716023@email.ncu.edu.cn (H.-Y.W.); 5701117111@email.ncu.edu.cn (G.-S.-L.Z.); 402432419003@email.ncu.edu.cn (H.-C.G.); dky@ncu.edu.cn (K.-Y.D.); 2School of Life and Science, Nanchang University, Nanchang 330031, China

**Keywords:** amniotic membrane, human amniotic stem cells, human amniotic mesenchymal stem cells, human amniotic epithelial stem cells, regenerative medicine

## Abstract

Stem cells including embryonic stem cells (ESCs), induced pluripotent stem cells (iPSCs) and adult stem cells (ASCs) are able to repair/replace damaged or degenerative tissues and improve functional recovery in experimental model and clinical trials. However, there are still many limitations and unresolved problems regarding stem cell therapy in terms of ethical barriers, immune rejection, tumorigenicity, and cell sources. By reviewing recent literatures and our related works, human amnion-derived stem cells (hADSCs) including human amniotic mesenchymal stem cells (hAMSCs) and human amniotic epithelial stem cells (hAESCs) have shown considerable advantages over other stem cells. In this review, we first described the biological characteristics and advantages of hADSCs, especially for their high pluripotency and immunomodulatory effects. Then, we summarized the therapeutic applications and recent progresses of hADSCs in treating various diseases for preclinical research and clinical trials. In addition, the possible mechanisms and the challenges of hADSCs applications have been also discussed. Finally, we highlighted the properties of hADSCs as a promising source of stem cells for cell therapy and regenerative medicine and pointed out the perspectives for the directions of hADSCs applications clinically.

## 1. Introduction

Stem cells, defined by dual hallmark features of self-renewal and differentiation potential, can be derived from embryonic and adult tissues. Stem cells are classified to pluripotent stem cells (PSCs), multipotent stem cells, and unipotent stem cells based on their developmental potency [1]. PSCs are able to form all tissues/cells with distinct functional properties which depend upon the derived and cultured conditions [2]. Embryonic stem cells (ESCs) and induced pluripotent stem cells (iPSCs) are the two most common types of PSCs [3]. Multipotent stem cells, such as hematopoietic stem cells, are restricted to generating the mature cell types of their tissue of origin and they exist in the resting state under normal physiologic circumstances and are activated when these tissues receive nociceptive stimulation [4]. Unipotent stem cells possess the capability of self-renewal and limited differentiation potential and only produce a single cell type. The most typical unipotent stem cells are spermatogonial stem cells, which can only differentiate into sperm [5]. In the early embryo, PSCs represent progenitors for all tissues while later in the development, tissue-restricted adult stem cells (ASCs), including multipotent stem cells and unipotent stem cells, give rise to cells with highly specialized functions. Unlike ESCs and iPSCs, tissue-restricted ASCs are limited in their potency to the cell types of the tissue in which they reside [6]. ASCs derived from different tissues showed an attractive application clinically due to their abilities to differentiate into a certain type or a designated type of specific cells and have little risk of tumorigenicity and immune rejection [6,7,8,9]. When tissues and organs are damaged, sufficient tissue-ASCs are essential in maintaining tissue regeneration and functional integrity.

Although researchers have made endless efforts to improve the technologies of ESC and iPSCs, there still are two prominent hardship, tumorigenicity and low survival rate of transplanted cells/tissues, leading to enormous challenges in clinical application [10,11]. In addition, the differentiation of ESCs and iPSCs to different cells is a stepwise process that is involved in a combination of transcription factors. During the in vitro inducing process, the cells generated from transdifferentiation of ESCs or iPSCs may not possess biological function. In addition, ASCs have also certain limitations, such as the limited pluripotency, the reduced numbers with aging and the ability of the restricted expansion in vitro. Some studies have showed that ASCs were not intrinsically immunoprivileged, and under appropriate conditions, allogeneic ASCs might also induce immune rejection of an allogeneic graft [12,13]. In addition, studies also showed that the gradual accumulation of genetic mutations in human ASCs during life were able to be transmitted to daughter cells and initiate tumorigenesis [14,15].

Human amnion-derived stem cells (hADSCs) including human amniotic epithelial stem cells (hAESCs) and human amniotic mesenchymal stem cells (hAMSCs) have the great advantages over other stem cells such as renewal, multi-differentiation potential, no-tumorigenicity, low/no immunogenicity, no ethical or legal concerns and their potent paracrine effects, especially immunomodulatory effects, making them have a promising source of stem cells for cell therapy in various diseases [16,17,18].

## 2. Characteristics of Human Amnion-Derived Stem Cells

### 2.1. Advantages of hADSCs over Other Stem Cells

Placenta is composed of the amnion member, chorionic plate, decidua basalis, chorionic villi, cotyledons/interuillous space, and placental septa (Figure 1A) [8,19]. Among these placental components, the amniotic membrane serves as a suitable raw materials for cell-based therapy due to the large number of cells [20]. The amniotic membrane is a transparent, smooth, avascular and single-layered thin membrane (about 100 μm) composed of epithelium and mesenchyme. The membrane covers the fetus and holds the amniotic fluid [21]. Generally, amnion membrane has five layers including epithelium, basement membrane, compact layer, fibroblast layer and spongy layer (Figure 1B) [22]. Epiblast-derived hAMSCs and hypoblast-derived hAESCs are two primary stem cell types in amniotic membrane which are responsible for the production of extracellular matrix (ECM), different cytokines and growth factors [8]. hAESCs come from the innermost layer of amnion which directly contact with amniotic fluid and fetus, whereas hAMSCs are scattered in the membrane [18]. Isolation protocols have been extensively described for both hAESCs and hAMSCs. Briefly, the amniotic membranes were treated with trypsin-EDTA for 45–60 min at 37 °C to release hAESCs [23]. Then, the remaining amniotic membranes were digested with Collagenase IV on a rotator 40 min at 37 °C to isolate hAMSCs [24]. 

hADSCs are easily isolated and propagated ex vivo. hAMSCs show a fibroblast-like morphology in culture [24,25], while hAESCs exhibit a cobblestone-like morphology [23]. Compared with stem cells from other sources, hADSCs have the following advantages: (1) Easy to obtain, abundant sources, and no ethical and moral disputes: as the remaining after fetal birth the amniotic membrane can be used for separation of hAMSCs and hAESCs, which will not harm the donors; (2) No tumorigenicity: numerous studies showed that hADSCs had no proliferation and growth on soft agar in vitro, no colonies formed, no teratoma formation after implanting NOD-SCID mice in vivo [24]; (3) Low immunogenicity and high histocompatibility: hADSCs were considered as the immune-privileged cells and showed remarkable characteristics of low immunogenicity [26,27]. hADSCs had a low expression of the major histocompatibility class I antigen (*HLA-ABC*), no expressions of the major histocompatibility class II antigen (*HLA-DR*) and β2 microglobulin [28,29,30,31], importantly, the cells did not express HLA-ABC costimulatory molecules such as CD80, CD86 and CD40 [32,33]. It has been reported that transplantation of hAMSCs into humans to treat lysosomal diseases showed no obvious rejection [34]. A recent study also demonstrated that intravenous administration of hAESCs did not result in haemolysis, allergic reactions, toxicity or tumor formation [35]. Akle et al. reported that an immunotype-mismatched human amniotic membrane did not elicit a host immune response when transplanted under the volunteers’ skin [36]. However, a few studies have highlighted that human amniotic cells might not actually be considered immune privileged, but, on the contrary, could stimulate both an innate and adaptive immune response, indicating that the possible co-immunostimulatory effects of amniotic stem cells [17,37].

### 2.2. The Molecular Markers of hADSCs

Both hAMSCs and hAESCs expressed the classical mesenchymal stem cells (MSCs) markers such as CD90, CD44, CD73, and CD105 [8], and lack of cell surface markers such as CD45, CD34, CD45, HLA-CR, CD80, CD86. The molecular markers of hAMSCs and hAESCs are shown in Table 1. Expressions of MSCs markers in hAMSCs indicated that the cells possess the attractive clinical benefits of MSCs due to immune-privilege and the ability for immunomodulation. 

### 2.3. High Pluripotency of hADSCs

hADSCs have the potential to differentiate into all three germ layers when exposed to exogenous growth factors or chemicals [36]. Both hAMSCs and hAESCs expressed the typical surface markers of embryonic stem cells such as SSEA-3, SSEA-4, SOX-2, TRA1-60 and TRA1-81 [51], especially the Oct-4 and Nanog [24], indicating the great potential of hAMSCs and hAESCs in regenerative medicine. So far, studies showed that hADSCs were able to differentiate into adipocytes [33,52], bone cells [34,38], nerve cells [39], cardiomyocytes [40,53], skeletal muscle cells [54,55], hepatocytes [34], hematopoietic cells [56], endothelial cells [57], kidney cells [58] and retinal cells [59,60] (Figure 2).

Although various therapeutic approaches have been applied to promote angiogenesis, most of them were not able to fully mimic the process of natural vessel development. Amnion has both angiogenic and anti-angiogenic properties, which is surface dependent. The epithelial surface of amnion had inhibitory effects on vessel formation [50] and hAESCs were able to incorporate into the arterial wall without immunosuppression, but failed to improve vascular function [61]. However, the vessel length and sprout were increased in the amniotic mesenchymal side [12]. hAMSCs shared similar capability with bone marrow MSCs in neovascularization [62] and could initiate the cascade of signals by secreting factors needed for promoting formation of stable neo-vasculature and angiogenesis [62]. Several studies have evaluated the proangiogenic potential of hAMSCs which expressed high levels of representative proangiogenic genes *VEGF-A*, *angiopoietin-1*, *HGF*, and *FGF-2* and anti-apoptotic factor *AKT-1*. By directly transplantation of hAMSCs into the ischemic hindlimbs of mice, the augmented blood perfusion and capillary density were observed, indicating that hAMSCs might promote the formation of neovascularization [63]. Moreover, hAMSCs exerted the beneficial paracrine effects on infarcted rat hearts through cardioprotection and angiogenesis [64].

hAMSCs and their conditional medium (hAMSC-CM) significantly improved the cutaneous blood flow after infusion into the ischemic leg whose femoral vessels were ligated [65]. König et al. observed that hAMSCs were able to take up acetylated low-density lipoprotein and form endothelial-like networks [66]. Studies showed that hAMSCs promoted angiogenesis by regulating the ERK1/2-MAPK signaling pathway [13] and that knockdown of lnRNA H19 significantly inhibited the angiogenic function of hAMSCs, in which the mechanism might be related to EZH2 degradation and VASH1 activation [67]. In addition, Tang et al. demonstrated that the angiogenesis of hAMSCs might be related to inhibit the functions of the Circ-ABCB10/miR-29b-3p/VEGFR, Circ-100290/miR-449a/eNOS, and Circ-100290/miR-449a/VEGFA axes in human umbilical vein endothelial cells [68,69]. Taken together, although the underlying mechanism of hAMSCs in angiogenesis is not fully elucidated the cells may become a useful reagent in various vascular diseases.

### 2.4. Immunosuppressive and Immunomodulatory Effects of hADSCs

hAMSCs-derived soluble factors including TGF-β, HGF, PGE2 and IDO suppress mitogen-induced peripheral blood mononuclear cell (PBMC) proliferation in a dose-dependent manner [70]. Wolbank et al. demonstrated the contact- and dose-dependent inhibitions of PBMC immune responses of hAMSCs and hAESCs [32]. Bulati et al. observed that INF-γ-induced immunomodulatory effects of hAMSC were dependent on the activated lymphomonocytes, cell-to-cell contact and soluble factors [71]. hAMSCs significantly inhibited the proliferation of stimulated lymphocytes and T cells [72]. Rossi et al. demonstrated that prostaglandins released by hAMSCs were responsible for the anti-proliferative effect on lymphocytes [70]. Meesuk et al. found hAMSCs exhibited immunosuppressive effect when they were co-cultured with activated T-cells by secreting indoleamine 2,3-dioxygenase [73]. Morandi et al. observed that HLA-G and -E molecules were involved in hAESCs-mediated suppression of T cell proliferation [74].

Furthermore, hADSCs also have immunomodulatory and immunosuppressive effects on inflammatory processes including reducing the activities of inflammatory cells and inhibiting migration of microglia and recruitment of immune cells to injury sites [16,17]. hADSCs also exhibited angiogenic, cytoprotective, anti-scarring, and antibacterial properties [75]. Therefore, it is reasonable to believe that hADSCs may be a potential cell source for cell-based therapy of diseases.

## 3. Cell-Based Therapy with Human Amnion-Derived Stem Cells

A number of preclinical studies successfully demonstrated that hADSCs hold the potential to alleviate many diseases (Figure 3).

### 3.1. Gynecological Diseases

Premature ovarian failure (POF) refers to amenorrhea before age 40 caused by ovarian failure. The primary or secondary amenorrhea is accompanied by the increase of gonadotropin level and the decrease of estrogen level, in turn, resulting in a series of low estrogen symptoms such as hot flashes, sweating, facial flashes and low libido and so on. Studies showed that hAMSCs protected chemotherapy-induced ovary damage by reducing granulosa cell (GC) apoptosis and promoting angiogenesis, cell proliferation, and gene expression through homing the stem cells to ovaries after intravenous injection and secreting cytokines including FGF2, IGF-1, HGF, and VEGF in POF rats [76,77,78]. Ding et al. reported that hAMSCs improved the proliferation and inhibited the apoptosis of ovarian GC cells from natural ovarian aging patients by secreting HGF and EGF [79]. hAMSCs restored the oxidation-induced damages of ovarian structure and function in POF mice by improving the local microenvironment of the ovary [80]. hAMSCs also exerted a higher therapeutic efficacy in cyclophosphamide induced ovarian insult in rats compared to the MSCs derived from adipose tissue [81]. Ling et al. found that hAMSCs treated by low-intensity pulsed ultrasound had more advantages for repairing ovarian injury and improving ovarian function in chemotherapy-induced primary ovarian insufficiency rats [82]. Zhang et al. demonstrated that hAESCs restored normal follicles in ovary, inhibited apoptosis of GCs by activating TGF-beta/Smad signaling pathway through paracrine effect [83,84]. Wang et al. observed that hAESCs were able to home into ovaries to differentiate into GCs [85]. By comparison of hAMSCs and hAESCs, Ding et al. found that hAMSCs were more effective in the treatment of POF, but hAESCs might cause immune rejection [37].

In addition, both hAMSCs and hAESCs have considerable potential for treatment of other gynecological diseases. Studies shown that both hAMSCs and hAESCs effectively treated intrauterine adhesion and promoted endometrial regeneration through paracrine effects [86,87,88]. Motedayyen et al. observed that hAESCs were able to treat unexplained recurrent spontaneous abortion through rescuing maternal immune abnormalities, inhibiting the activation of T cells and reducing the production of the activated proinflammatory cytokines [89].

### 3.2. Nervous System Diseases

Spinal Cord Injury (SCI) is one of severe central nervous system diseases caused by the dysfunction of motor and sensory [90]. To date, no effective treatment is satisfactory for the recovery of this type of neurological disease. Zhou et al. observed that there was a neuroprotection in SCI rats by injections of hAMSCs nearby the injury site and intravenous transplantation and the possible mechanism might be related with promoting the regeneration of axons and angiogenesis [91]. Sankar et al. reported that hAESCs graft prevented the formation of glial scars in the transection lesion site and inhibited the apoptosis of axotomized neurons [92]. In addition, studies showed that grafted hAESCs promoted the axon regeneration and sprouting, and also prevented the atrophy of axotomized red nuclei in SCI rats [93] and reduced the SCI-induced mechanical allodynia [94]. And with co-treatment of melatonin and Wnt-4, hAESCs might transform into neural cells and join tissue repair [95]. These studies indicate that hAESCs and hAMSCs might facilitate the recovery of SCI.

Alzheimer’s Disease (AD) is the most common chronic neurodegenerative disease as showing extensive degeneration in a variety types of neurons [96]. Studies showed that hAMSCs transplantation might alleviate the pathological alternation and facilitate the restoration of the memory functions in AD mice by attenuating oxidative stress [97] or breaking Aβ deposition [98]. Similar with the study, transplantation of hAESCs into the lateral ventricles of APP/PS1 AD mice increased the numbers of cholinergic neurons and acetylcholine concentration in hippocampus [99]. Moreover, mice with engrafted hAESCs displayed a significant improvement in spatial memory [99,100]. Microarray analysis showed that hAESCs possessed a higher capability to treat AD with verbenalin [101].

Autism spectrum disorder (ASD) is a complicated disorder in neural development involved in social connectivity and interaction [102]. Zhang et al. observed that the intraventricular injection of hAESCs alleviated the social deficits by rescuing the function of hippocampus in BTBR mice [103]. Mucopolysaccharidosis type 1 (MPS1) is characterized by the dysfunction of lysosomal storage and is often accompanied by neurological degeneration [104] and hAESCs injection ameliorated the ability of sensorimotor coordination in MPS1 mice [105]. Peripheral nerve injury is a grievous peripheral nervous system disease which leads to the loss of function entirely [106] and Li et al. reported that hAMSCs-derived some neurotrophic factors contributed to peripheral nerve repair [107]. Parkinson’s disease (PD) is a neurodegenerative disorder characterized by a progressive death of nigral dopamine neurons with a resultant loss of striatal dopamine levels and study showed that hAESCs were beneficial to 6-OHDA-induced PD rats by secreting the neurotrophic factors [108].

### 3.3. Brain Injury

Stroke or cerebral apoplexy is a server debilitating disease that is characterized by symptoms of hemorrhagic injury and brain ischemia [15]. Since there was no effective treatment for stroke and cerebral apoplexy, stem cell-based treatment has been explored for a long time [109]. Recent studies have demonstrated that hAESCs exhibited a neural differentiation and neuroprotection effect by expressing melatonin receptor MT1 in vitro model of stroke [110] and systemic administration of hAESCs reduced brain inflammation, infarct development, and functional deficits in poststroke patients [111]. Numerous studies showed that hAMSCs and the conditional medium of hAMSCs (hAMSCs-CM) have a neuroprotection in the phase of cerebral ischemia [112,113], in which the possible mechanism might be related to restoring mTOR activity and inhibiting autophagy [114] or activating neuronal ERK1/ERK2-BDNF signaling pathway [115]. It has been reported that hAESCs significantly ameliorated behavioral dysfunction and reduce infarct volume of ischemic rats [116]. However, Tao et al. observed that the alleviation of hAMSCs transfected with brain derived neurotrophic factor (BDNF) gene (*BDNF*-hAMSCs) on the behavioral dysfunction of cerebral ischemia rats was more rapid and effective compared with rats treated with hAMSCs only [117].

Intracerebral hemorrhage (ICH) is caused by rupture of intracranial blood vessels, which is accounting for about 10–15% of all strokes. hAMSCs treatment significantly promoted neurological recovery in rats after ICH [118] and attenuated the inflammation and promoted the remyelination in autoimmune encephalomyelitis (EAE) mice [119].

Perinatal brain injury is highly associated with neurodevelopmental disability after preterm birth and the intrauterine inflammation is a common cause to the injury. hAESCs were able to reduce inflammation-induced fetal brain injury in the preterm ovine fetus [120] and improved white matter maturation after asphyxia in preterm fetal sheep [121]. Leaw et al. observed that hAESCs-derived trophic factors directly rescued cell death by immunomodulating brain microglia [122]. Elisa et al. found that hAMSCs protected the brain acute injury by secreting metabolites rather than proteins or ribonucleic molecules [123]. These results demonstrated that hAESCs may afford a therapeutic utility in the management of perinatal brain injury.

### 3.4. Lung Diseases

Studies showed that hAMSCs reduced lung injury, lung fibrosis, CT score and inflammation when the cells were injected into white smoke inhalation-induced mice [124] and the atomization of hAMSCs offered an effective way to deliver the cells to the airway for treatment of lung injury [125]. Tan et al. observed that hAESCs resulted in significant changes of T-cells, macrophages, dendritic cells, and monocyte/macrophage infiltration along with the elevation of lipoxin-A4 [126].

Particularly, bronchopulmonary dysplasia (BPD) is a chronic lung disease that mainly affects premature babies who require ventilator support [127]. With a two-year infant-enrolled study, Malhotra et al. reported that low-dose allogeneic hAESCs were safe to the vulnerable premature infants with BPD [127,128]. Zhu et al. also observed that early hAESCs treatment was more efficacious than late treatment in reducing interleukin-1β, TNF-α and monocyte chemoattractant protein-1 levels [129]. Likewise, hAESCs derived from term-pregnancy were able to activate bronchioalveolar stem cells and type 2 alveolar epithelial cells compared to preterm-pregnancy hAESCs following bleomycin challenge [130]. At the genetic level, Li et al. observed that hAMSCs ameliorated hyperoxia-induced neonatal lung injury with a significant increased expression of aminoacyl-peptide hydrolase [131]. In addition, epithelial disruption by polidocanol provided a new effective way for intratracheal transplantation of human amniotic stem cells [132].

As an end-stage change in some lung diseases, pulmonary fibrosis refers to the scarring formation in lungs caused by abnormal repair in the damaged lung tissue. By injecting hAESCs into bleomycin-induced SCID mice with lung fibrosis, Moodley et al. found that lung collagen contents were notably decreased, indicating that there was a great improvement of lung fibrosis with hAESCs treatment [133]. Subsequently, Carbone et al. observed that hAMSCs were be able to successfully induce the increasing of F-actin content as well as the reorganization of stress fibers and tight junctions by co-cultured with cystic fibrosis (CF) respiratory epithelial cells in purpose of correcting basic defects associated with CF [134]. In addition, hAESCs-derived exosomes played a crucial role in lung repair by modulating inflammatory and fibrogenic pathways [135] and the therapeutic effects were enhanced by combining with the anti-fibrotic drug, serelaxin [136]. Zhang et al. also found that hAMSCs with overexpressed *Nrf2* further reduced lung injury, lung fibrosis, and inflammation in LPS-challenged mice [137].

Growing evidence suggests a mechanistic link between inflammation and pulmonary fibrosis as well as lung injury [138,139]. Studies showed that hAESCs were be able to attenuate the fetal pulmonary inflammatory response to experimental intrauterine inflammation and reduce consequent alterations in lung development [139] and furthermore, hAMSCs were showed to slow down the process of pulmonary fibrosis by reducing B-cell response which was induced in the chronicity of lung inflammatory processes [138]. Geng et al. found that early hAESCs intervention was an effective way to delay disease progression in rats with chronic obstructive pulmonary disease (COPD) [140].

### 3.5. Liver Diseases

Acute liver injury (ALI) refers to abnormal liver function caused by various reasons. Studies showed that pretreatment with hAMSCs partially protected the acetaminophen-induced ALI by inhibiting Kupffer cells (KCs)-related innate immune inflammation and KCs autophagy [141] and that hAESCs and hAESC-CM significantly reduced hepatic inflammation and fibrosis in a high fat diet-induced non-alcoholic fatty liver disease model by inhibiting pSMAD 2/3 signaling and reducing the numbers of activated hepatic stellate cells and liver macrophages [142]. Furthermore, hepatocyte transplantation to treat liver disease is largely limited by the availability of the hepatocyte-like cells (HLCs). hAESCs possess the ability to differentiate into HLCs with similar functions to human primary hepatocytes in vitro [143,144] and in vivo [145]. We also observed that transplantation of HLCs via the tail vein protected mice from CCL4-induced ALI [23]. Obviously, transplantation of HLCs provides a new treatment for ALI.

Liver fibrosis, the precursor to cirrhosis, is a complex inflammatory and fibrogenic condition caused by chronic liver injury and an imbalance in extracellular matrix (ECM) synthesis and degradation mediated primarily by activated hepatic stellate cells (HSCs) [146]. hAMSCs and hAMSC-CM were observed to inhibit the activation of HSCs and in turn, to improve liver fibrosis along with reductions of fibrosis markers α-smooth muscle actin (α-SMA), platelet-derived growth factor (PDGF) and collagen I [147,148] and the infiltration of KC cells [149]. hAMSC-CM have the ability to upregulate the expressions of ECM degradation-related genes (*MMP-2*, *MMP-9*, *MMP-13*, and *TIMP-1*) and proliferation-related gene cyclin B which could suppress HSC proliferation. These findings provided a plausible explanation for the protective role of hAMSCs in liver fibrosis [150]. Lee et al. reported that CD34-positive hAMSCs might be a suitable candidate with a better safety for cell-based therapy in treatment of liver fibrosis [151].

Liver cirrhosis is the sixth leading cause of death in adults, mainly due to constantly liver cell damage and continuous and unregulated healing of liver wounds [152]. The transplantation of hAMSCs significantly decreased the fibrosis formation and progression of CCl4-induced cirrhosis [153] and improved the liver functions by reducing hepatic inflammation and oxidative stress in the cirrhotic rats [154]. In addition, it has been reported that hAESCs were be able to differentiate into functional hepatocytes in the mouse cirrhosis model [155].

Maple syrup urine disease (MSUD) is an inborn error of metabolism characterized by elevated branched-chain keto acids and branched-chain amino acids such as leucine, isoleucine, valine resulting in severe brain injury and death unless treated. hAESCs transplantation lengthened survival and corrected many amino acid imbalances in a mouse model of MSUD [156,157], suggesting that hAESCs were deserved for treatment of metabolic liver diseases clinically.

### 3.6. Cancer

Although a great progress has been made in the treatment of cancers, the morbidity and mortality still stay at a high level [158]. It has been reported that hAMSCs were able to home to tumor sites and inhibit the growth of human hepatoma cells by promoting cell apoptosis and inhibiting cell proliferation [159]. We demonstrated that the mechanisms of hAMSCs’ anti-hepatocellular carcinoma were mainly involved in their paracrine effects, in which hAMSC-derived DKK-3, DKK-1 and IGFBP-3 markedly inhibited cell proliferation and promoted apoptosis of Hepg2 cells through suppressing the Wnt/β-catenin signaling pathway and IGF-1R-mediated PI3K/AKT signaling pathway, respectively [25]. hAMSCs also significantly reduce the proliferation of cancer cell lines of hematopoietic [160] and prostate cancer [161] by inducing cell cycle arrest and inducing C6 glioma apoptosis through the Bcl-2/Caspase pathways [162]. Contrary to previous reports, hAMSC-CM significantly enhanced the proliferation of lung carcinoma cells in a 3D microfluidic tumor model [163]. In addition, Bonomi et al. demonstrated that hAMSCs were able to use as vehicles for the delivery of cytotoxic agents to inhibit tumor cell proliferation [164].

Coincidentally, hAESCs and hAESC-CM also showed an anti-angiogenic, pro-apoptotic, and anti-proliferation effect in cancer cells [165,166]. Bu et al. showed that hAESCs endowed the potential anticancer properties on epithelial ovarian cancer which is partially mediated by hAESC-secreted TGF-β1-induced cell cycle arrest [167]. Vaccination of mice with hAESCs led to expansion of systemic and splenic cytotoxic T cell population and induction of cross-protective cytotoxic responses against tumor cells, resulting in a significant reduction in colon cancer tumor volume [146].

### 3.7. Skin Diseases

Both hAMSCs and hAESCs possess considerable therapeutic potential for diabetic wounds through secreting angiogenic factors (IGF-1, EGF, and IL-8) and enhancing re-epithelialization and cellularity [22,168]. Our results showed that hAMSCs and hAMSCs-CM efficiently cured heat stress-induced skin injury by inhibiting apoptosis and promoting proliferation of skin cells through activating PI3K/Akt signaling pathway [24]. Gao et al. observed that the exosome-derived miR-135a from hAMSCs promoted cutaneous wound healing and fibroblast migration by downregulating *LATS2* levels [169]. A combined administration of hAMSCs/Matriderm was beneficial to potentiate the therapeutic effects of hAMSCs on wound healing [170].

hAESCs also possess therapeutic capability for wound healing through facilitating migration and proliferation of keratinocytes [171] and increasing cellularity and re-epithelialization with a paracrine mechanism, which might be mediated by ERK, JNK and AKT signaling pathways [172]. Zhou et al. reported that hAESCs significantly improved the wound healing by upregulating the proangiogenic factor VEGF and downregulating the inflammatory cytokine TNF-α [173]. hAESCs-derived exosomes promoted the migration and proliferation of fibroblasts and accelerated wound healing, which might be related with stimulating the expression of MMP-1 [174]. Besides paracrine effects, the air-liquid interface stimulated early differentiation of organotypic hAESCs to epidermal cells (skin-like substitute) [175] and reconstruct tissue-engineered (TE) skin in organotypic culture [176]. Yu et al. establish a new model for reconstruction of bilayer TE skin with hAMSCs and hAESCs [177] and the TE skin was similar in morphology to human skin, in which they had stratified epidermis and underlying dermis and successfully repaired full thickness skin defects [178]. Although both hAMSC-CM and hAESC-CM have been proved to promote wound healing, the levels of wound healing related proteins such as CTHRC1, LOXL2, and LGALS1 in hAMSC-CM were significantly higher than those in hAESC-CM [179].

hAMSCs and hAESCs also treated other skin diseases such as keloid, skin aging, and psoriasis. hAMSC-CM prevented the proliferation and activation of keloid fibroblasts [180] and hAESC-CM attenuated TGF-β1-induced human dermal fibroblast transformation to myofibroblasts via TGF-β1/Smad3 pathway [181], suggesting that hAESC-secreted cytokines might promise for keloid treatment as a topical agent. Reports also showed that hAMSCs delayed the oxidative stress-induced skin aging by paracrine action [182,183]. Imai et al. observed that hAMSCs suppressed the development of psoriasiform dermatitis and the keratinocyte response to pro-inflammatory cytokines in a mouse model [184].

### 3.8. Bone Diseases

Osteoarthritis (OA) is a disease of the synovial joint marked by chronic, low-grade inflammation leading to cartilage destruction. hAMSCs significantly ameliorated severity of arthritis and decreased the histopathological changes by their anti-oxidation and anti-inflammation in collagen-induced arthritis rats [185]. Compared with human adipose stem cells, hAMSCs possessed more chondroprotective by improving OA chondrocyte viability, increasing cartilage glycosaminoglycan content and reducing OA synovial macrophage M1:M2 ratio [186]. hAMSCs also attenuated diabetes-induced dysfunction of bone marrow mesenchymal stem cells (BM-MSCs) through promoting p38 mitogen-activated protein kinase and VEGF-mediated antioxidant activity [187]. Studies showed that hAMSCs promoted implant osseointegration and bone regeneration by accelerating mineralized deposition rates on implant surfaces and in bone-augmented areas [188] and enhanced new bone formation by inducing M2 macrophages to secrete BMP-2 and VEGF in rabbits’ skull defects [189]. In addition, hAESCs promoted osteoblasts differentiation through the secretion of TGFβ1 and miR-34a-5p [190] and increased the chondrocytes in the injured area to accelerate the repair of osteochondral defect in rabbit [191]. All these results indicated that both hAMSCs and hAESCs might be an optimal cell source for bone regeneration.

Although the most properties of hAMSCs and hAESCs were similar in terms of their morphology, proliferation, immunophenotype profile, and osteogenic capacity [192], hAMSCs were often used for the treatment of bone diseases. hAMSCs co-cultured with 3D silk fibroin scaffolds greatly enhanced osteogenic differentiation and angiogenesis in vitro and in vivo [193]. Using human amniotic membrane (HAM) as scaffold, López et al. demonstrated that HAM was a useful material for articular cartilage repair, in which in combination with hAMSCs, HAM showed better potential for cartilage repair with similar reparation capacity than chondrocytes [194]. hAMSCs transfected with FGF-2 combined with the human acellular amniotic membrane scaffold could accelerate tendon-to-bone healing in a rabbit extra-articular model [195].

### 3.9. Cardiovascular Disease

It has been reported that hAMSCs significantly reduced the size of atherosclerotic lesions by inhibiting immune response via NF-κB pathway in mouse model [196] and improved left ventricular function, increased capillary density and the levels of angiogenic cytokine, angiopoetin (Ang)-1 and vascular endothelial growth factor (VEGF-A) in myocardial infarction [197,198] and decreased myocardial fibrosis area in ischemic heart tissue [199]. Studies showed that hAMCs exhibited antiapoptotic and proangiogenic functions in myocardial infarction of rat heart through secreting small extracellular vesicles [200] and were be able to differentiate into cardiomyocyte-like cells, decreased infarct size, and improved cardiac function in vivo [201]. Naseroleslami et al. found that in the presence of magnetic field, the superparamagnetic iron oxide nanoparticles-labeled hAMSCs significantly improved cardiac function and reduced fibrosis and tissue damage by suppressing inflammation in a NF-κB/MAPK-dependent mechanism [202]. In addition, the transplantation of hAMSCs in rats with heart failure not only decreased the level of fibrosis but also conferred significant improvement in heart performance in terms of echocardiographic and hemodynamic parameters [203]. Maleki et al. also demonstrated that intravenous injection of hAMSC-CM has a therapeutic effect on heart failure by reducing fibrosis and increasing levels of angiogenesis [204].

### 3.10. Inflammatory Bowel Diseases

Crohn’s disease (CD) and Ulcerative Colitis (UC) are two main inflammatory bowel diseases (IBD) which are characterized by chronic, relapsing, inflammatory conditions of the gastrointestinal tract. hAMSCs and hAMSC-CM significantly decreased the infiltration of neutrophils and monocytes/macrophages and the expressions of *TNF-α, CXCL1*, and *CCL2* in rat colitis models induced by 2,4,6-trinitrobenzene sulfonic acid (TNBS) [205]. Similar to the above results, hAMSCs and hAMSC-CM were able to ameliorate dextran sulphate sodium-induced colitis in mice via a paracrine anti-inflammatory effects [206], possibly the mechanism might be related to inhibiting the activities of monocytes/macrophages and the activation of NF-kB signaling pathway [207]. hAMSCs also improved radiation proctitis induced by therapeutic irradiation for intra-pelvic cancer often causes radiation proctitis, possibly through inhibiting inflammatory reactions [208]. More recently, Otagiri et al. designed a phase I/II clinical trial to investigate the safety and efficacy of hAMSCs in patients with treatment-resistant Crohn’s disease [209].

### 3.11. Kidney Disease

Injection of hAMSCs in the chronic kidney failure (CKF) mice significantly decreased the levels of serum creatinine and urea, and improved fibrosis in kidney [210]. Ren et al. found that hAESCs improved mortality and renal function in acute kidney injury (AKI) mice by decreasing the number of apoptotic cells, preventing peritubular capillary loss, and modulating kidney local immune response, in which the mechanism might be related to alleviating renal ischemia reperfusion injury by hAESCs-derived exosomes [211]. Liu et al. demonstrated that hAESCs-mediated reduction of c-Rel expression and in turn, a reduction in IL-2 expression and release were achieved by stimulating miR-101 expression in AKI patient-derived CD4 + T cells [212].

### 3.12. Diabetes Mellitus

Diabetes mellitus (DM) is a chronic disease that results in a variety of systemic complications [213]. The most common forms of diabetes mellitus are type 1 diabetes, in which there is an absolute deficiency of insulin due to the damage of pancreatic beta cell destruction, and type 2 diabetes mellitus, in which insulin resistance may lead to hyperglycemia [214]. hAESCs had multidirectional differentiation potential similar to embryonic stem cells and pluripotent stem cells [116]. Luo et al. observed that hAESCs transplantation significantly reduced hyperglycemia symptoms, increased the plasma insulin content, and partially repaired the islet structure in type 1 diabetic mice and the combination of hAESCs with hyaluronic acid exhibited a remarkable therapeutic effect compared to hAESCs alone group, indicating that hyaluronic acid was an efficient co-inducer of the differentiation of hAESCs into functional insulin-producing cells [215]. Peng et al. found that the combination of nicotinamide plus betacellulin promoted the differentiation of hAESCs into islet-like cell clusters by reducing the DNA methylation [1]. Zou et al. observed that miR-32 effectively inhibited the expression of *WWP2* of hAESCs and promoted the cells were differentiated into β islet-like cells, demonstrating that the overexpression of *Oct4* in hAESCs was better to maintain the pluripotency [216]. They also observed that lncRNA-ROR effectively maintained Sox2 gene expression through competitively binding to miR-145, achieving pluripotency maintenance in hAESCs and regulation of their directed β islet-like cell differentiation efficiency [5]. In addition, Okere et al. demonstrated for the first time that hAESCs cultured in 3D serum-free conditions could functionally produce newly synthesized insulin and successfully respond to glucose stimulation [217] and hAMSCs also could be differentiated into functional insulin-secreting cells which normalized the hyperglycemia in type 1 diabetic mice [218].

Organoids play an important role in basic research, drug screening, and regenerative medicine. In 2019, the viable and functional islet organoids were successfully generated from dissociated islet cells (ICs) and hAESCs through 3D co-culture. The integration of hAESCs into the islet cell structure significantly improved the functionality, viability and engraftment capability of islet organoids through multiple mechanisms, including better resistance to ischemia, accelerated revascularization and restoration of cell-to-matrix contacts [219].

### 3.13. Autoimmune Diseases

Multiple sclerosis (MS) is an autoimmune inflammatory disease of the central nervous system. In vitro, hAESCs suppressed both specific and non-specific T cell proliferation, decreased pro-inflammatory cytokine production, and inhibited the activation of stimulated T cells. T cell responses and production of the pro-inflammatory cytokine interleukin (IL)-17A were reduced in hAESCs-treated experimental autoimmune encephalomyelitis (MS-like disease) mice [220]. Liu et al. observed that hAESCs significantly suppressed splenocyte proliferation and reduced central nervous system CD3 + T cell and F4/80+ monocyte/macrophage infiltration and demyelination by secreting TGF-β and PGE2 [221]. All these results demonstrated that hAESCs hold promise for treatment of autoimmune diseases like MS.

The chronic inflammation of autoimmune diseases such as experimental autoimmune uveitis (EAU), hashimoto’s thyroiditis (HT), and systemic lupus erythematosus (SLE) develops repetitive localized destruction or systemic disorders. hAESCs ameliorated the pathological progression of EAU and preserved the retinal structure organization and thickness by downregulating T helper (Th)17 cells and upregulating T regulatory (Treg) cells [222] and played a role in their inflammation inhibition and injury recovery in HT and SLE murine models [223]. Pre-eclampsia (PE) is one of the most severe syndromes in human pregnancy. It has been reported that PE-hAMSCs inhibited CD4/CD8 T-cell proliferation and Th1/Th2/Th17 polarization, induced Treg to block dendritic cells and M1 differentiation, switched them to M2 cells, suggesting that there was no intrinsic impairment of the immunomodulatory features of PE-hAMSCs [224]. Compared with 2D cultures, hAMSC spheroids kept in 3D culture system remained viable and secreted more angiogenic and immunosuppressive factors [225].

All these results demonstrated that hAMSCs-mediated inhibition of inflammatory processes was achieved by inhibiting the activities of inflammatory cells and the recruitment of certain immune cells to injury sites through their immunomodulatory and immunosuppressive effects.

### 3.14. Other Diseases

Niemann–Pick disease type C1 (NPC) is an autosomal recessive cholesterol storage disorder characterized by liver dysfunction, hepatosplenomegaly and progressive neurodegeneration. hAESCs have the ability to extend the life span, reduce the rapid loss of weight, and alleviate tissue damage of NPC mice [226]. hAMSCs and hAMSC-CM ameliorated biliary hyperplasia, peribiliary fibrosis, and inflammation in a rat model of sclerosing cholangitis, indicating that hAMSCs might represent a new modality for treating sclerosing cholangitis [46]. Muscular atrophy is characterized by the volumetric muscle loss (VML) with at least 20% or more. Zhang et al. observed that hAMSCs increased angiogenesis and improved local tissue repair with expressing skeletal muscle specific markers when the cells were implanted into rat tibialis anterior muscle with VML, demonstrating that hAMSCs were differentiated into skeletal muscle [55]. Navas et al. found that intracameral injection of hAMSCs promotes corneal wound healing by inducing an anti-inflammatory and anti-fibrotic environment [227]. Urethral defects caused by urethral trauma, congenital malformation, and tumor are common causes of urological surgeries. hAMSCs promoted urethral defect repair in rabbits [228] and restored spermatogenesis in a busulfan-induced testis toxicity mouse model by resisting apoptosis and oxidative stress [229], suggesting that hAMSCs might represent a promising tool for urethral repair and spermatogenesis.

## 4. Clinical Trials with hAMSCs and hAESCs

In the 1980s and 1990s, hAESCs had been transplanted in volunteers and patients with lysosomal storage diseases [34,47] or Sphingomyelinase Deficiency [34]. All the clinic trials showed that there were no acute immune rejection and tumor formation after the transplantation of hAESCs. In addition, hAESCs showed no differences in global histone acetylation or mitochondrial point mutation accumulation and a decreased global DNA methylation compared to dermal fibroblasts [43], indicating that hAESCs were genetically stable and were one of the safest stem cell sources for cell-base therapy clinically. In recent years, several clinical trials have been conducted for evaluating the safety and efficacy of hADSCs in a variety of diseases. In 2018, Lim et al. undertook a first-in-human clinical trial to evaluate the safety of hAESCs in babies with BPD, in which allogeneic hAESCs (1 × 10^6^/ kg) were intravenously infused to six premature babies, and the results showed that there were no adverse events including immunological rejection, indicating that allogeneic hAESCs were safe for the babies with BPD [127]. Two years later, they reported that one infant died in the neonatal period due to the unrelated causes and the five surviving infants were alive without long-term adverse events including tumor formation during the 2-years observation of the hAESCs clinical trial [128]. At present, phase I cell dose escalation study was applied to evaluate the safety and efficacy of intravenous hAESCs infusions in 24 preterm infants at high risk of severe BPD [230]. Furthermore, the safety of intravenously delivered allogeneic hAESCs was evaluated in patients with compensated liver cirrhosis [152] and stroke patients [231], respectively. In addition, Otagiri et al. assessed the safety and efficacy of hAMSCs in patients with treatment-resistant Crohn’s disease [209]. Certainly, these open label studies would provide solid data of the phases 2 and 3 clinical trials to achieve the ultimate goal of developing hAESCs as a therapeutic option for patients with different diseases.

In October 2020, there was one clinical trial of “amniotic mesenchymal stem cells” listed in the clinical trial database (ClinicalTrials.gov), therefore, so far, a total of 12 clinical trials have been proposed (Table 2) for allogeneic hADSCs applications including lupus nephritis, Spastic Cerebral Palsy, Nonunion Fracture, Corneal Epithelial Dystrophy, Bronchial Fistula, Primary Ovarian Insufficiency, Acute-graft-versus-host Disease, Asherman’s Syndrome, Intrauterine Adhesion, Leukemia, and Parkinson’s Disease (Figure 4). Obviously, the outcomes of these clinical trials will provide evidence for hADSCs applications clinically once the safety and efficacy of the cells are confirmed.

## 5. hADSCs Clinical Applications

### 5.1. The Possible Therapeutic Mechanisms of hADSCs in Treating Diseases

By reviewing hADSCs treating 36 diseases in 12 major systemic disorders above, we found that except for the trans-differentiation toward the damaged cells, inhibiting apoptosis and promoting the regeneration, the paracrine effects-related immunosuppressive and immunostimulatory features of hADSCs were play a key role in treating various diseases, in which hADSCs were able to secret a variety of growth factors including angio-modulatory cytokines, anti-bacterial peptides, and anti-inflammatory agents, exhibiting their angiogenic, cytoprotective, immunosuppressive, anti-inflammatory, anti-scarring, and antibacterial properties. Actually, the paracrine effects of hADSCs have been considered as the main underlying mechanisms in the cells-based therapy, in which the possible therapeutic mechanisms of hADSCs were summarized in Table 3.

### 5.2. The Challenges of hADSCs Applications

Recently, hADSCs have been applied for treating certain diseases clinically. However, there are still some bottlenecks and limitations, which hinder the clinical application of hADSCs. In general, the characterization of hADSCs can be affected by the gestational age, region of cell isolation in placenta, isolation protocols, cross-contamination, passage numbers, and measuring methods etc. Some studies revealed that primary and expanded hADSCs displayed different characteristics and paracrine effects. Besides, there are many challenges for hADSCs applications in pre-clinical and clinical trials. (1) Cell quality: highly purified hADSCs with high viability have to be prepared in the laboratory with good manufacturing procedure (GMP) requirements in which standard manufacturing and cryopreservation process should be established. (2) Safety: for clinical trials, the primary consideration is safety. The biggest challenges are whether allogeneic hADSCs can be well tolerated in human and how to avoid acute adverse events related to hADSCs during and after the administration. The possible adverse events including local site reaction, anaphylaxis, infection, features of rejection, and tumor formation should be closely observed. (3) Efficacy: another challenge in clinical trial is effectiveness. The phenotype and the biological properties of hADSCs are significantly affected by some factors and cells. Most of the preclinical research of hADSCs were conducted in mice, which are vastly different from humans. Therefore, it is logically to assume that, when the cells are injected into human, an extremely complex microenvironment, the behaviors and characteristics of hADSCs and hADSC-derived conditional medium could not be dramatically changed in an unpredictable way. In some cases, to validate the efficacy of hADSCs the preclinical studies should be carried out in primates before the cells are implemented in human.

Therefore, establishing a reasonable and optimal isolation protocol and agreeing upon strict definitions for hADSCs will be necessary for their future application in clinical trials. In addition, some studies reported that hADSCs have different or even opposite effects in the treatment of some diseases, such as cancer. In some diseases, there is an exacerbation of inflammatory conditions that need to be dampened, but in other diseases, the stimulation of immune system has been proposed as an efficient therapeutic strategy. In order to improve therapeutic outcome, the mechanisms for the therapeutic potential of hADSCs need to be carefully elucidated. Moreover, although numerous studies indicated that there is no tumorigenicity and low/no immunogenicity for hADSCs, more preclinical animal studies should be conducted to further confirm these features before they are applied for treating various diseases clinically.

## 6. Conclusions and Perspectives

Human amnion-derived stem cells (hADSCs) including hAMSCs and hAESCs have the great advantages over other stem cells in team of abundant sources, no ethical and moral disputes, no tumorigenicity and low/no immunogenicity. Moreover, the pluripotent properties and paracrine effects of hADSCs significantly enhanced their applications in experimental research and clinical practice. All these properties make them a promising source of stem cells for cell therapy and regenerative medicine.

Over the past decades, mounting evidences have demonstrated the efficacy of hADSCs therapies in vitro and in vivo. More recently, numerous clinical trials have been conducting throughout the world for evaluating the safety and efficacy of hADSCs in different diseases. Obviously, the outcomes of these clinical trials will provide evidence for hADSCs applications clinically once the safety and efficacy of the cells are confirmed. There is no doubt that the stem cells including hADSCs-based therapies will hold a promise for treatments of a wide range of diseases.

In perspective, more studies are needed to further investigate the pluripotent properties and elucidate the mechanisms of paracrine effects. Accumulating evidence demonstrated that hADSCs provided a beneficial microenvironment for cell survival by activating endogenous mechanisms of tissue regeneration through secreting bioactive cytokines and exosomes. This was supported by evidence that the injection of the conditioned media (CM) or exosomes of hADSCs could achieve a positive outcome similar to that of cell transplantation, representing an acceptable alternative for stem cell-free biotherapy. hADSC-derived cytokines and exosomes have an important potential for repairments in various diseases, avoiding the shortcomings of stem cell transplantation. Thus, exploring and identifying the most effective secretome components, including bioactive factors and exosomes secreted by hADSCs, will provide new treatment strategies for regenerative medicine in the future. Besides paracrine effects, hADSCs also have strong capability to transdifferentiate into cardiomyocytes, hepatocyte, epidermal cells, corneal epithelial-like cells, neural cells, and functional insulin-secreting cells in vitro following chemical induction, biological treatment, gene transfection, or coculture with other types of cells, suggesting that in the future, inducing the differentiation of hADSCs toward a desired phenotype in vitro before transplantation will be an effective method for treatment of heart disease, liver injured, diabetes and so on. In addition, the more wide clinical trials of hAMSCs and hAESCs in treatment of different diseases should be conducted to successfully achieve their applications clinically.

## Figures and Tables

**Figure 1 ijms-22-00970-f001:**
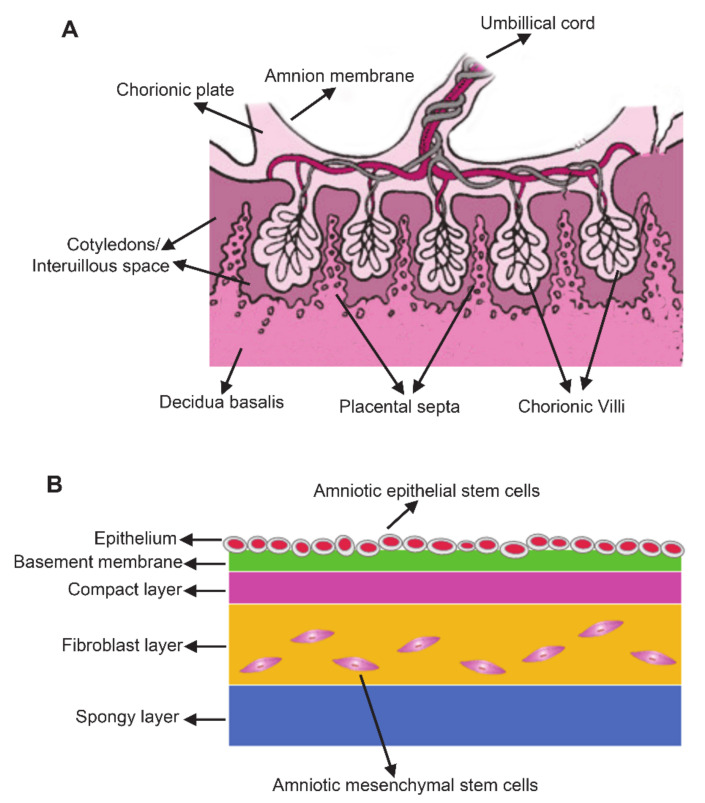
The anatomy of the human term placenta and amniotic membrane. (**A**) Schematic section of the human term placenta; (**B**) Schematic of amnion structure.

**Figure 2 ijms-22-00970-f002:**
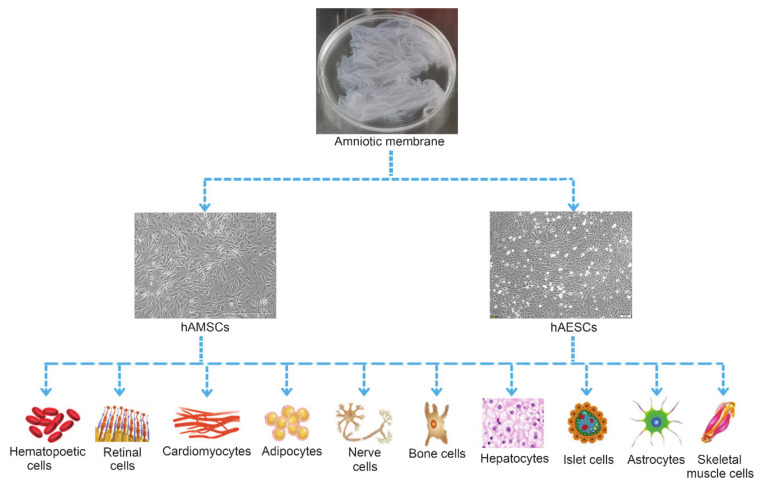
Multi-differentiation potential of hAMSCs and hAESCs.

**Figure 3 ijms-22-00970-f003:**
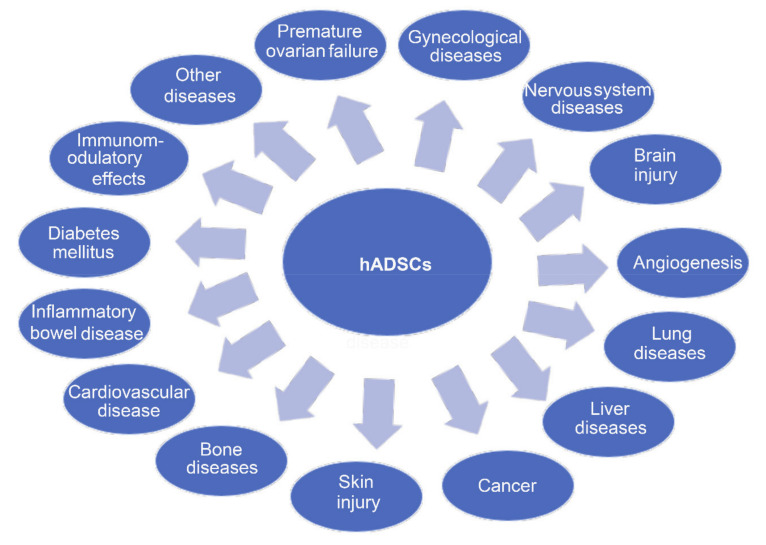
Diseases have been treated by hADSCs.

**Figure 4 ijms-22-00970-f004:**
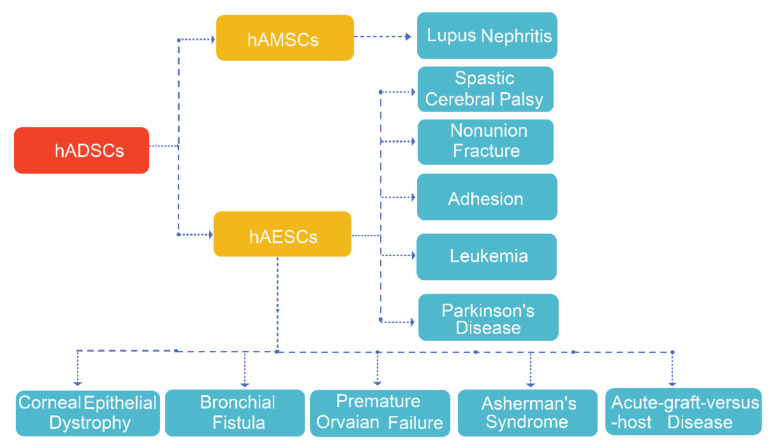
Clinical application of hADSCs.

**Table 1 ijms-22-00970-t001:** Expression of markers of human amniotic mesenchymal stem cells and amniotic epithelial stem cells.

hADSCs	Epithelial Markers	Mesenchymal Stem Cell Markers	Pluripotent Markers	Hematopoietic Marker	The Major Histocompatibility Complex and Their Co-Stimulatory Molecules		References
	Positive	Positive	Positive	Negative	Positive	Negative	
**hAESCs**	Cytokeratin, E-cadherin, CD49f, CD326,	CD29, CD166,	OCT4, NANOG, SSEA4, TRA-1-60	CD34, CD45		HLA-DR, HLA-DQ	Yang et al. [35]
	CK19	CD29, CD44, CD73, CD90, CD105,	SSEA-4, SOX2, OCT-4	CD31, CD34, CD45, CD49d		HLA-DR	Wu et al. [38]
	CK7, E-cadherin	CD29, CD73, CD105	OCT4, NANOG, SSEA4	CD34, CD45	HLA-ABC	HLA-DR, CD80, CD86, CD40	Liu et al. [23]
	E-cadherin		OCT4, SOX2, NANOG, TFE3, KLF4, SSEA3, SSEA4, TRA-1-60, REX1				Castro et al. [39]
	E-cadherin, CD49f, CK7, EpCAM	CD44, CD90, CD105, CD146, PDGFR-b, CD29		CD45	HLA-A-B-C	HLA-DP-DQ-DR, CD80, CD86, CD40	Pratama et al. [33]
	Cytokeratin		SSEA3, SSEA4, TRA-1-60, Oct-4	CD34			Evron et al. [40]
		CD29, CD73		CD34, CD45	HLA-ABC	HLA-A2, HLA-DQ, HLA-DR	Murphy et al. [30]
		CD9, CD10, CD29, CD104, CD49f, CD105, CD44		CD34, CD45	HLA-ABC	HLA-DR, CD80, CD86, CD40	Banas et al. [31]
			SSEA-3, SSEA-4, TRA 1-60, TRA 1-81	CD34			Miki et al. [41]
	Cytokeratin, E-cadherin	CD29, CD166, CD90,	OCT4, NANOG, TRA 1-60, SOX2	CD34, CD45		HLA-DR, HLA-DQ	Yang et al. [35]
	CK19, E-cadherin	CD29, CD44, CD90	OCT4, SOX2, SSEA4,	CD34			Wu et al. [42]
	E-cadherin,	CD9, CD24,	SSEA-3, SSEA-4, TRA 1-60, TRA 1-81, Oct-4, Nanog	CD34			Miki et al. [36]
		CD73, CD29,	Oct3/4, Sox2, Klf4, SSEA4, c-Myc.	CD34, CD45			Koike et al. [43]
**hAMSCs**		CD29, CD44, CD49d, CD73, CD90, CD105	SSEA-4, SOX2, OCT-4	CD31, CD34, CD45		HLA-DR	Wu et al. [38]
		CD29, CD73, CD90, CD105	OCT4, NANOG, SSEA4	CD34, CD45, CD133	HLA-ABC	HLA-DR, CD80, CD86, CD40	Liu and Li et al. [24,25]
		CD44, CD90, CD105, CD146	Oct-3/4	CD45, CD34	HLA-ABC	HLA-DR	Bačenková et al. [44]
		CD90, CD44, CD73, CD166, CD105, CD29	SSEA-4, STRO-1	CD34, CD45			Prado and Sugiura et al. [45,46]
		CD29, CD44, CD73, CD90, CD105	SSEA-4, Oct4	CD34, CD45, CD133	HLA-ABC	HLA-DR	Coppi et al. [34]
		CD105, CD117,		CD34	HLA-ABC	HLA-DR	Borghesi et al. [47]
		CD29, CD105	Oct-3/4, SSEA-4, SOX2, NANOG, Rex-1		HLA-A, HLA-DQB1		Mihu et al. [48]
		CD44, CD90, CD105. CD90		CD31, CD34			Seo et al. [49]
		CD44, CD73, CD90, CD105, CD29, CD49f, CD271	Oct3/4, Sox2, Klf4,SSEA4, Nanog, TRA1-60	CD34, CD45			Koike et al. [43]
		CD44, CD73, CD90, CD105, Vimentin	OCT3/4, C-MYC, SOX2, NANOG, SSEA-3, SSEA-4	CD34, CD45		HLA-DR	Nogami et al. [50]

**Table 2 ijms-22-00970-t002:** Clinical trials using hAMSCs and hAESCs as a biological intervention.

	Title	NCT Number	Condition or Disease	Phase	Enrollment	Injection Site	Dose	Outcome Measures	Status	Locations
hAMSCs	Allogeneic Amniotic Mesenchymal Stem Cell Therapy for Lupus Nephritis	NCT04318600	Lupus Nephritis	Phase 1	16	Peripheral intravenous	1 × 10^6^/kg	Incidence of Adverse Events; 2. Changes in 24h urine protein quantification before and after treatment; 3. Changes in eGFR before and after treatment; 4. Changes in SLEDAI score before and after treatment.	Completed	China
hAESCs	Effect of Human Amniotic Epithelial Cells on Children With Spastic Cerebral Palsy	NCT03107975	Spastic Cerebral Palsy	Phase 1	10			Gross Motor Function Measure-66; 2. Fine Motor Function Measure; 3. Modified Ashworth Scale; 4. Gesell Developmental Scales.	Unknown	China
	Treatment of Nonunion of Limb Fracture With Human Amniotic Epithelial Cells	NCT03031509	Nonunion Fracture	Phase 1 Phase 2	36	Nonunion site	50 million	time for bone union; 2. pain in fracture site; 3. limb functional score	Not yet recruiting	China
	Transplantation of Tissue Cultured Human Amniotic Epithelial Cells Onto Damaged Ocular Surfaces	NCT00344708	Corneal Epithelial Dystrophy		0			Ocular surface healing.	Not Applicable	United States
	Human Amniotic Epithelial Cells for Treatment of Bronchial Fistula	NCT02959333	Bronchial Fistula	Phase 1	10		3–5 × 10^7^/5 mL	median time for fistula closure; 2. Complication	Unknown	China
	A Therapeutic Trial of Human Amniotic Epithelial Cells Transplantation for Primary Ovarian Insufficiency Patients	NCT02912104	Primary Ovarian Insufficiency;Premature Ovarian Failure;Infertility	Phase 1	2	Bilateral ovarian artery infusion	2 × 10^7^/5 mL	bilateral ovarian artery infusion; 2. Hormonal Assessment serum estrogen level will be measured; 3. Hormonal Assessment serum AMH level will be measured; 4. Pregnancy rate will be measured.	Recruiting	China
	Human Amniotic Epithelial Cells Prevent Acute Graft-versus-host Disease After Hematopoietic Stem Cell Transplantation	NCT03764228	Acute-graft-versus-host Disease	Not Applicable	27		1 × 10^6^, 2 × 10^6^, 5 × 10^6^ cell/kg	Occurrence of adverse events; 2. Occurrence of Graft-versus-host disease after hAECs infusion	Recruiting	China
	Human Amniotic Epithelial Cells Treatment for Ovarian Insufficiency	NCT03207412	Premature Ovarian Failure	Not Applicable	20	bilateral ovarian tissue	200 million	Changes of follicle stimulating hormone; 2. antral follicle count	Unknown	China
	Human Amniotic Epithelial Cells for Asherman’s Syndrome	NCT03223454	Asherman’s Syndrome	Phase 1	50	uterine cavity	100 million	Menstrual blood volume; 2. Endometrial thickness; 3. Uterine volume; 4. Ongoing pregnancy rate	Not yet recruiting	China
	Human Amniotic Epithelial Cell in Treatment of Refractory Severe Intrauterine Adhesion	NCT03381807	Intrauterine Adhesion	Early Phase 1	20	uterine cavity	100 million	changes of Endometrial thickness; 2. changes of Menstrual blood volume; 3. pregnancy rate	Not yet recruiting	China
	hAECs Are Preliminarily Applied in Allogeneic Hematopoietic Stem Cell Transplantation	NCT03759899	Leukemia		30			Cell counts of T cells, NK cells, B cells and monocytes.	Recruiting	China
	Stereotactic Transplantation of hAESCs for Parkinson’s Disease	NCT04414813	Parkinson’s Disease	Early Phase 1	3	Stereotactic	50 million	Number of Participants with adverse event and serious adverse event; 2. Changes in Unified Parkinson’s Disease Rating Scale; 3. Changes in the Hoehn and Yahr scales; 4. Changes in Parkinson’s Disease Questionnaire; 5. Changes in the Schwab and England scores.	Not yet recruiting	China

**Table 3 ijms-22-00970-t003:** Biological functions of hAMSCs and hAESCs in various diseases.

Diseases		Biological Functions of hAESCs and hAMSCs	References
Gynecological diseases	Premature ovarian failure	granulosa cells (GCs) apoptosis promoting angiogenesis and proliferation to restoring the oxidation-induced damages secreting cytokines to improving the local microenvironment rescuing maternal immune abnormalities inhibiting the activation of T cells and reducing the production of the activated proinflammatory cytokines	[76,77,78]
	Intrauterine adhesion		[86,87,88]
	Recurrent spontaneous abortion		[89]
Nervous system diseases	Spinal Cord Injury	promoting the regeneration of axons and angiogenesis preventing the formation of glial scars, so promoting the axon regeneration and sprouting secreting the neurotrophic factors exhibiting a neural differentiation and neuroprotection effect reducing brain inflammation, so attenuating oxidative stress attenuating the inflammation by immunomodulating brain microglia	[91,92,93,94,95]
	Alzheimer’s Disease		[97,98,99,100,101]
	Autism spectrum disorder		[24,107]
	Parkinson’s disease		[108]
	Stroke		[109,110,111,112,113,114,115,116,117]
	Intracerebral hemorrhage		[118,119]
	Perinatal brain injury		[120,123]
Lung diseases	bronchopulmonary dysplasia	Reducing inflammatory factors by modulating inflammation and reducing B-cell response Ameliorating hyperoxia-induced neonatal lung injury Promoting repair of lung injury, and activating bronchioalveolar stem cells Inhibiting fibrogenic pathways	[133,134,138]
	pulmonary fibrosis		[133,134,135,136,137]
Liver diseases	Acute liver injury	Inhibiting innate immune inflammation differentiating into HLCs, and inhibiting the activation of HSCs, so reducing hepatic fibrosis reducing oxidative stress, and protecting liver function	[23,141,142,143,144,145]
	Liver fibrosis		[147,148]
	Liver cirrhosis		[153,154,155]
Cancer	hepatocellular carcinoma	homing to tumor sites and inhibit the growth of human tumor cells by promoting cell apoptosis and inhibiting cell proliferation led to expansion of systemic and splenic cytotoxic T cell population and induction of cross-protective cytotoxic responses Improving tumor microenvironment, inhibiting inflammatory mediators for promoting tumor growth	[25,159]
	Hematopoietic cancer		[160]
	prostate cancer		[161]
	Glioma		[162]
	epithelial ovarian cancer		[167]
	colon cancer		[146]
Skin diseases	skin injury	inhibiting apoptosis and promoting proliferation of skin cells upregulating the proangiogenic factor VEGF and downregulating the inflammatory cytokines stimulating early differentiation of organotypic hAESCs to epidermal cells	[24,171,179]
	Keloid		[180,181,182,183,184]
Bone diseases	Osteoarthritis	anti-oxidation and anti-inflammation, reducing OA synovial macrophage M1:M2 ratio promoted osteoblasts differentiation, increased the chondrocytes growth improving OA chondrocyte viability, increasing cartilage glycosaminogly	[185,186,187,188,189,190,191]
Cardiovascular disease	myocardial infarction	reducing the size of atherosclerotic lesions by inhibiting immune response improving ventricular function, increased capillary density and the levels of angiogenic cytokine improving cardiac function and reduced fibrosis and tissue damage by suppressing inflammation	[200,201,202]
	heart failure		[203,204]
Inflammatory bowel diseases	Crohn’s disease	decreasing the infiltration of neutrophils and monocytes/ macrophages inhibiting the expressions of inflammatory factors ameliorating dextran sulphate sodium-induced colitis	[205,206,207]
	Ulcerative Colitis		[206,207,208,209]
Kidney disease	chronic kidney failure	modulating kidney local immune response, and reductiing IL-2 expression alleviating renal ischemia reperfusion injury	[210,211,212]
Diabetes mellitus	type 1 diabetes	differentiating into functional insulin-producing cells inhibiting the inflammatory response by diabetes reducing hyperglycemia symptoms, increased the plasma insulin content	[1,214,218]
	type 2 diabetes		[1,214,218]
Autoimmune diseases	Multiple sclerosis	suppressing both specific and non-specific T cell proliferation decreasing pro-inflammatory cytokine production inhibiting the activation of stimulated T cells downregulating T helper (Th)17 cells and upregulating T regulatory (Treg) cells	[220,221]
	experimental autoimmune uveitis		[222]
	hashimoto’s thyroiditis		[223]
	systemic lupus erythematosus		[223,224]
	Pre-eclampsia		[224]

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
