# Peer review of "Characteristics and Therapeutic Potential of Human Amnion-Derived Stem Cells"

_ijms, 2021, doi:10.3390/ijms22020970_

Round 1

Reviewer 1 Report

In this manuscript, the authors summarized the application of human amnion-derived stem cells (hASCs), including human amniotic mesenchymal stem cells (hAMSCs) and human amniotic epithelial stem cells (hAESCs). The authors first introduced human amniotic stem cells' characteristics, including sources, pluripotent markers, and major histocompatibility complex and their co-stimulatory molecules. The rest of the manuscript mainly focused on the current progress of hASCs studies and their potential therapeutic applications for pre-clinical and clinical research. Overall, this review has covered most of the recent studies regarding the applications of hASCs in different types of diseases. There are several issues that the authors need to address before publication.

Major issue:
1. The review article misses a paragraph of discussion of challenges of using hASCs in pre-clinical and clinical trials. The authors just listed the different applications for in vitro, in vivo, and clinical studies. But there is no connection between laboratory results and clinical trial results. It would be beneficial for the readers to understand the challenges of using hASCs for different applications in pre-clinical and clinical trials.
2. The abstract did not reflect the content of the manuscript. The authors should revise the abstract to reflect the content of the review.
For example, the abstract should not include the details of the characteristics of hASCs. Instead, the abstract should summarize the main content of the article.
3. In perspective, only two sentences say more studies are needed to investigate the pluripotent properties and mechanics’ of paracrine effects. More information should be included in this paragraph, i.e., the authors should give future perspectives of the usage and directions of hASCs for different diseases.

Minor issue:
1. Figure 1: are these images original? If not, the authors should get authorization from the original authors. The authors should not use any of these images without permission. I suggest the authors generate their own figures.

Reviewer 2 Report

The manuscript by Quan-Wen Liu et al, deals with the characterization and therapeutic potential of human aminiotic stem cells.
Unfortunately, in the introduction the authors confuse various classifications regarding the differentiation capacity and the associated potency of the cells.
For example, the introduction refers to the totipotency of pluripotent stem cells (p1 l.37). Also the description of the reprogramming according to Yamanaka is little
described scientifically, since no transcription factors were transferred (p2 l50), but the corresponding genes were introduced into the cells by retroviral gene transfer into the host cells. The inaccurate and partly wrong scientific way of expression runs through the whole introduction. On p2l557 ...without tumorgenecity and immune reaction (Citation #6)
nothing of the sort is claimed. Also the claim that hAMSC's have the mentioned advantages (p2 l66-68) is not supported with citations.

In the characterizing part of the hAMSC's, Table 1 refers to the surface markers OCT4, SOX2, Nanog and KLF4 as well as CK19. As described in the introduction, these are nuclear transcription factors
and cytoskeletal proteins, respectively. Therefore, the characterization part is also not scientifically tenable.

In the following part, the cell based therapy, a long list of different publications begins. Apart from the fact that these are almost exclusively in the mouse model and partly with rather questionable controls (PBS as control for transplantations instead of fibroblasts), no connection becomes clear to the reader.
A contextual connection is not apparent. The actual clinical therapies have not yet been completed for the most part, yet the authors speak to the fact that these approaches apparently will confirm safety and efficiacy (p 19 l522-523). This is also an absolutely untenable statement, as it cannot be supported with any evidence.

At the end, the authors repeat almost verbatim their introduction, followed by the untenable statement at the end of the clinical trials. A real conclusion that this resource perhaps also critically questions or with other resources critically argues is not to recognize.

Due to the listed inadequacies, I can not agree to a publication in its current form.

Reviewer 3 Report

Liu et al offered an overview of the potential, studies, and clinical application of amniotic stem cells.

The topic of this review is of potential interest for a large readership due to the characteristics of these cells.

However, the introductory part relies on an inappropriate description of stem cells. Concepts and classifications are often not correct. For example, potency is not defined correctly, nor the cells included in each related sub-classification. No mention of the difficulty to reach a differentiated functional phenotype is introduced regarding the use of pluripotent stem cells, especially. Actually, this is one of the main concerns with the application of all of these cells.

  • Lines 55-62: Adult stem cells can be immunogenic and induce tumorigenesis depending upon the considered type.
  • Lines 70-71: the addition of a figure depicting the characteristics of the amniotic membrane and the cells derived from each region will be beneficial for the readership. Moreover, methods applied to derive each cell line should be briefly reported, and eventually, methodical differences and their significance discussed.
  • Lines 82-83: Cases of tumor formation involving infused/injected amniotic stem cells have been reported.
  • Lines 88-89: The possible co-immunostimulatory effects of amniotic stem cells have been reported.
  • Lines 96-97: it is improper to state the potential of a cell line only by the expression of stem cell markers.
  • Table 1 misses several important studies regarding amniotic stem cells.
  • The part describing results from the application in cell therapy and clinical trials is very interesting. The reviewer suggests to introduce some tables and/or figures for each application to ease the readership.

There is an important use of acronyms, which sometimes may be confounding (e.g., ASCs for adult stem cells and hASCs for human amniotic fluid stem cells). The reviewer recommends using acronyms in a more consistent way.

A critical review of the literature would be beneficial to the readership. Are all these cells used in the different studies derived with similar protocols? Are they used plain or conditioned? Are there differences reported in the outcomes of applied amniotic stem cells for a given treatment? The reviewer feels that a methodological part should be introduced to comment on published results more critically.
